# Preoperative and operation-related risk factors for postoperative nosocomial infections in pediatric patients: A retrospective cohort study

Kuanrong Li[1‡], Xiaojun Li[1‡], Wenyue Si[1], Yanqin Cui[2], Huimin Xia[3], Xin Sun[4], Xingrong Song[5], Huiying Liang[1]*

1 Institute of Pediatrics, Guangzhou Women and Children's Medical Center, Guangzhou Medical University, Guangzhou, China, 2 Cardiac Intensive Care Unit, Heart Center, Guangzhou Women and Children's Medical Center, Guangzhou Medical University, Guangzhou, China, 3 Guangdong Province Key Laboratory of Structural Birth Defects, Guangzhou Women and Children's Medical Center, Guangzhou Medical University, Guangzhou, China, 4 Department of Medical Administration, Guangzhou Women and Children's Medical Center, Guangzhou Medical University, Guangzhou, China, 5 Department of Anesthesiology, Guangzhou Women and Children's Medical Center, Guangzhou Medical University, Guangzhou, Guangdong, China

‡ These authors are co-first authors on this work.
* lianghuiying@hotmail.com

**Data Availability Statement:** Data cannot be shared publicly because of the institution's data protection rule against any leakage of confidential personal information. However, data might be

## Abstract

### Background

Pediatric patients undergoing invasive operations bear extra risk of developing nosocomial infections (NIs). However, epidemiological evidence of the underlying risk factors, which is needed for early prevention, remains limited.

### Methods

Using data from the electronic medical records and the NI reporting system of a tertiary pediatric hospital, we conducted a retrospective analysis to identify preoperative and operation-related risk factors for postoperative NIs. Multivariable accelerated failure time models were fitted to select independent risk factors. The performance of these factors in risk stratification was examined by comparing the empirical risks between the model-defined low- and high-risk groups.

### Results

A total of 18,314 children undergoing invasive operations were included for analysis. After a follow-up period of 154,700 patient-days, 847 postoperative NIs were diagnosed. The highest postoperative NI rate was observed for operations on hemic and lymphatic system. Surgical site infections were the NI type showing the highest overall risk; however, patients were more likely to develop urinary tract infections in the first postoperative week. Older age, higher weight-for-height z-score, longer preoperative ICU stay, preoperative enteral nutrition, same-day antibiotic prophylaxis, and higher hemoglobin level were associated with delayed occurrence of postoperative NIs, while longer preoperative hospitalization,

available from the institution's ethics committee (contact via z_simian@hotmail.com) for researchers who are considered eligible to have access, and the eligibility will be assessed by the institution's ethics committee on a case-by-case basis.

**Funding:** The authors received no specific funding for this work.

**Competing interests:** The authors have declared that no competing interests exist.

**Abbreviations:** AFT, accelerated failure time; ASA, American Society of Anesthesiologists; CVC, central venous catheterization; EN, enteral nutrition; EMR, electronic medical record; HR, hazard ratio; ICD-9-CM, International Classification of Diseases, 9th Revision, Clinical Modification; ICU, intensive care unit; NI, nosocomial infection; PD, patient-day; PN, parenteral nutrition; SSI, surgical site infection; SWC, surgical wound classification; TR, time ratio; UC, urinary catheterization; UTI, urinary tract infection; WAZ, weight-for-age z-score; WBC, white blood cell.

longer operative duration, and higher American Society of Anesthesiologists score showed acceleration effects. Risk stratification based on these factors in an independent patient population was moderate, resulting in a high-risk group in which 72% of the postoperative NIs were included.

## Conclusions

Our findings suggest that pediatric patients undergoing invasive operations and at high risk of developing postoperative NIs are likely to be identified using basic preoperative and operation-related risk factors, which together might lead to moderately accurate risk stratification but still provide valuable information to guide early and judicious prevention.

## Introduction

Nosocomial infections (NIs) pose a long-standing challenge to clinical practitioners and remain one of the leading causes of in-hospital mortality [1]. The overall prevalence of NIs varies from 7% in affluent countries to 15% in economically developing countries [2, 3], whereas the most common types of NIs are invariably surgical site infections (SSIs) and device-associated infections [2, 4, 5], suggesting that most of the NIs are attributable to invasive operations.

Pediatric patients undergoing invasive operations face extra risk of developing NIs because of their underdeveloped immune system. According to two European studies, the NI incidence was 2.5% in general pediatric wards and was 17% in surgical wards [6, 7]. From a prevention perspective, by recognizing the preoperative and operation-related factors that enhance patients' susceptibility to postoperative NIs, healthcare providers would be able to identify vulnerable patients for closer observation and to initiate timely prophylactic treatment when necessary. For adult surgical patients, several epidemiological studies have revealed a wide variety of such factors [8–10], but detailed research in pediatric patients is scarce.

In the present study, we focused on pediatric patients undergoing invasive operations and aimed to identify preoperative and operation-related risk factors for the occurrence of postoperative NIs.

## Methods

### Study setting and design

The present study was a hospital-based retrospective cohort study conducted in a tertiary referral hospital—Guangzhou Women and Children's Medical Center—in Guangzhou, China. The data source of this study was the electronic medical records (EMRs) of the pediatric inpatients who underwent invasive procedures between 2016 and 2018.

### Extraction of data and ascertainment of study outcomes

Clinical data were derived from the EMRs and then linked to the hospital's NI reporting system via patient identification numbers. In the EMRs, operative procedures were documented in both text and the procedural codes defined by the International Classification of Diseases, 9th Revision, Clinical Modification (ICD-9-CM). According to the criteria of the NI reporting system, an infection was considered nosocomial if it occurred > 48 hours after admission. Neonatal infections were considered nosocomial as well if they were acquired during delivery. Diagnoses of specific NI types were made following the criteria from the Centers for Disease

Control and Prevention/National Healthcare Safety Network [11]. The outcome of interest in the present study was NI diagnosed after invasive operation. For patients with multiple invasive operations and/or multiple postoperative NI episodes, only the first invasive operation and the first postoperative NI were analyzed.

## The patient cohort

In the EMR database, we identified 18,314 patients who underwent invasive operations between 2016 and 2018 on one of the following specific systems: the nervous system (ICD-9-CM code: 01–05), respiratory system (ICD-9-CM code: 30–34), cardiovascular system (ICD-9-CM code: 35–39), hemic and lymphatic system (ICD-9-CM code: 40–41), digestive system (ICD-9-CM code: 42–54), urinary system (ICD-9-CM code: 55–59), and musculoskeletal system (ICD-9-CM code: 76–84). The main reason for us to focus on these systems was that the invasive operations performed on these systems except those on the hemic and lymphatic system were the most common ones in our institution. Invasive operations on the hemic and lymphatic system were relatively rare but were included because of the high postoperative NI rate after hemic and lymphatic surgeries.

## Ethical considerations

This study was approved by the ethics committee of the Guangzhou Women and Children's Medical Center (2019–13600). This study utilized historical data collected during routine clinical practice and the findings of this study will be used only for academic activities; therefore requirement for informed consent from patients was waived.

## Statistical analysis

The following candidate risk factors were considered: sex; age, nutritional status measured with weight-for-age z-score (WAZ), and blood test result at admission; lengths of preoperative hospitalization and intensive care unit (ICU) stay; preoperative enteral nutrition (EN) and parenteral nutrition (PN) support; operative duration; surgical implantation; antibiotic prophylaxis; the American Society of Anesthesiologists (ASA) score; surgical wound classification (SWC); and ICD-9-CM code. For patients younger than 10 years of age, WAZ was calculated using the WHO growth standards as the reference. For patients above that age, WAZ is not an appropriate measure of nutritional status and thus was treated as missing. Antibiotic prophylaxis was defined as use of antibiotics on the same day as the operation was performed. In order to retain the patients with missing values in analysis, binary indicators were created to denote data incompleteness, which was the case for WAZ, operative duration, and ASA score. Binary indicators were also created for preoperative ICU, blood test at admission, and surgical incision to distinguish between patients who did and who did not receive these procedures (see Table A in S1 File for a detailed description).

Time to event was defined as the duration from date of operation to date of NI diagnosis or date of discharge, whichever came first. After confirming that the log-transformed time to event was approximately normally distributed, accelerated failure time (AFT) models with log-normal distribution were fitted. In log-normal AFT models, the exponentiated coefficient ($e^{\beta}$) of a factor is interpreted as time ratio (TR) indicating whether this factor would decelerate ($e^{\beta} > 1$) or accelerate ($e^{\beta} < 1$) the occurrence of the event. The TR estimates can be converted into the number of days by which the time between operation and occurrence of postoperative NIs could be prolonged or shortened using the following formula: $\Delta T = T_{ref} \{exp[\beta_x(x-x_{ref})] -1\}$, where $T_{ref}$ denotes the time between the operation and the occurrence of postoperative NI for a reference patient, and $X_{ref}$ denotes the risk factor profile of the reference patient.

A reduced model was achieved using backward elimination with a threshold *P*-value of 0.1. Indicator variable and the variables related to it were handled as a block during variable selection; therefore we chose a less stringent *P*-value to decrease the possibility of excluding blocks that contain statistically significant risk factors. In order to control for the clinical heterogeneity of the operations, we predetermined that ICD-9-CM code should be included in the models regardless of the result of variable selection. We also built multivariable Cox regression models to estimate the hazard ratios (HRs), which are more commonly used to describe the strengths of risk associations.

In order to examine the clinical applicability of the identified risk factors for risk stratification purpose, we derived a postoperative NI risk score for another data set of 4,383 pediatric patients who underwent invasive operations between January 2019 and May 2019, of them 110 NIs were diagnosed postoperatively. The postoperative NI risk score was calculated by summing up the risk factors weighted by the AFT model coefficients and used the median of this risk score as a cutoff to divide the patients into low- and high-risk groups (Risk score calculation in S1 File). We compared the two groups regarding their empirical postoperative NI risks, which were estimated using the Nelson-Aalen method [12].

All the statistical tests were two-sided with $P < 0.05$ considered statistically significant. All the statistical analyses were performed using the "survival" package in R (R Foundation for Statistical Computing, Vienna, Austria).

## Results

Baseline characteristics of the cohort, stratified by the subsequent NI status, are shown in Table 1. Compared with the patients who did not develop postoperative NIs, those who did were relatively younger, had a lower WAZ and longer preoperative hospitalization, and were more likely to have preoperative ICU stay and to receive preoperative nutrition support. The NI group had on average a lower hemoglobin level. Regarding operation-related characteristics, patients in the NI group had an averagely longer operative duration and were more likely to receive surgical implantation and to have higher ASA scores.

During a follow-up of 154,700 patient-days, 847 postoperative NIs were diagnosed, yielding an incidence rate of 5.5 per 1000 PDs. Overall, operations on hemic and lymphatic system were followed by the highest NI rate (11.0 per 1,000 PDs), and SSIs were the most common NI type (Table 2). However, when confined to the first postoperative week, the highest NI risk was observed for cardiovascular operations (Fig 1A), and the NI type with the highest risk was urinary tract infections (UTIs, Fig 1B).

In the full multivariable AFT model, older age, higher WAZ, longer preoperative ICU stay, preoperative EN, antibiotic prophylaxis, and higher hemoglobin level were statistically significantly associated with delayed occurrence of postoperative NIs (Table 3). Longer preoperative hospitalization, blood test at admission, and longer operative duration were statistically significantly associated with advanced occurrence of the events. Backward elimination led to a reduced model including all the prior statistically significant risk factors as well as high ASA score ($\geq$ III). According to the reduced AFT model and given a reference patient who was defined as follows: underwent an invasive operation on the hemic and lymphatic system, age = 12 months, WAZ = −3, days of preoperative hospitalization = 7, preoperative ICU = 0, preoperative EN = 0, preoperative antibiotic prophylaxis = 0, hemoglobin = 100g/L, WBC = $10 \times 10^9$/L, operative duration = 120 minutes, and ASA score = II, the estimated time between invasive operation and the occurrence of postoperative NIs was 56 days. Therefore, antibiotic prophylaxis = 1, preoperative EN = 1, and one year older in age would extend this time by approximately 24, 14, and 4 days, respectively, for otherwise comparable patients; whereas

**Table 1. Baseline characteristics of a retrospective cohort of pediatric patients who underwent invasive operations (n = 18,314), stratified by postoperative NI status, the Guangzhou Women and Children's Medical Center, 2016–2018.**

| | Patients without postoperative NI (n = 17,467) | Patients with postoperative NI (n = 847) |
|---|---|---|
| Sex, male (%) | 12,208 (64.9) | 559 (63.9) |
| Age in months, median (IQR) | 23 (6–59) | 16 (3–47) |
| WAZ, indicator: yes (%) | 16,292 (93.3) | 795 (93.9) |
| WAZ median (IQR) [a] | −0.64 (−1.57–0.11) | −0.93 (−2.02–−0.03) |
| Preoperative hospitalization days, median (IQR) | 2 (1–5) | 3 (1–6) |
| Preoperative ICU stay, indicator: yes (%) | 6,559 (37.6) | 382 (45.1) |
| Preoperative ICU days, median (IQR) [a] | 2 (1–4) | 2 (1–6) |
| Preoperative EN, yes (%) | 4,233 (24.2) | 237 (28.0) |
| Preoperative PN, yes (%) | 1,576 (9.0) | 118 (13.9) |
| Antibiotic prophylaxis, yes (%) | 11,400 (65.3) | 521 (61.5) |
| Preoperative blood tests, indicator: yes (%) | 15,039 (86.1) | 765 (90.3) |
| Hemoglobin (g/L), median (IQR) [a] | 117 (104–127) | 109 (94–124) |
| WBC ($10^9$/L), median (IQR) [a] | 9.5 (7.1–12.4) | 9.2 (6.2–13.3) |
| Surgical implantation, yes (%) | 2,926 (16.8) | 175 (20.7) |
| Operative duration, indicator: yes (%) | 14,481 (82.9) | 515 (60.8) |
| Operative duration (min), median (IQR) [a] | 130 (80–190) | 175 (100–260) |
| ASA score, indicator: yes (%) | 14,728 (84.3) | 597 (70.5) |
| ASA score I, n (%) [a] | 3,277 (22.2) | 75 (12.6) |
| ASA score II, n (%) [a] | 8,896 (60.4) | 280 (46.9) |
| ASA score ≥III, n (%) [a] | 2,555 (17.4) | 242 (40.5) |
| SWC, indicator: yes (%) | 8,242 (47.2) | 310 (36.6) |
| Clean (%) [a] | 6,302 (76.5) | 229 (73.9) |
| Clean-contaminated (%) [a] | 1,271 (15.4) | 57 (18.4) |
| Contaminated (%) [a] | 669 (8.1) | 24 (7.7) |

[a]Limited to the patients for whom the data were available or applicable. ASA (American Society of Anesthesiologists), CVC (central venous catheterization), EN (enteral nutrition), ICU (intensive care unit), IQR (interquartile range), NI (nosocomial infection), PN (parenteral nutrition*SD* standard deviation), UC (urinary catheterization), WAZ(weight-for-age z-score), WBC *(*white blood cell).

preoperative hospital stay for one more week, one-point increase in ASA score, and one-hour increase in operation duration would advance the occurrence of the postoperative NIs by 11, 10 and 5 days, respectively.

The multivariable Cox model (Table B in S1 File) yielded largely consistent results except for the statistically significant positive association for WBC. In the reduced Cox model, the proportional hazards assumption was not met for age at operation, antibiotic prophylaxis, and operative duration.

Risk stratification based on the reduced AFT model was applied to the patients who underwent invasive operations between January 2019 and May 2019 (n = 4,383). Fig 2 shows the empirical postoperative NI risks in the resulting low- and high-risk groups. Within the first 30 days after operation, the high-risk group (2,191 patients including 79 NIs) showed a consistently higher postoperative NI risk than the low-risk group (2,192 patients including 31 NIs). The 30-day postoperative NI rates were 3.1 and 5.4 per 1,000 PDs for the two groups, respectively, and 72% of the postoperative NIs occurred in the high-risk group.

**Table 2. Incidence of major NIs in a retrospective pediatric patient cohort (n = 18,314) after invasive operations, by operative site (ICD-9-CM) and by NI type, the Guangzhou Women and Children's Medical Center, 2016–2018.**

| | Patients | NIs (rate, per 1,000 PDs) | Specific NI types | | | | | | | |
|---|---|---|---|---|---|---|---|---|---|---|
| | | | SSI | UTI | URI | LRI[a] | GI | BSI | Others | Unknown |
| **Total** | 18,314 | 847 (5.5) | 182 | 148 | 122 | 115 | 78 | 63 | 83 | 56 |
| **By operative site (ICD-9-CM)** | | | | | | | | | | |
| Nervous system (01–05) | 2,353 | 182 (7.6) | 81 | 15 | 12 | 15 | 5 | 10 | 33 | 11 |
| Respiratory system (30–34) | 1,875 | 74 (4.4) | 13 | 4 | 24 | 22 | 6 | 3 | 2 | 0 |
| Cardiovascular system (35–39) | 3,233 | 209 (6.4) | 20 | 55 | 23 | 45 | 20 | 10 | 11 | 25 |
| Hemic and lymphatic system (40–41) | 466 | 81 (11.0) | 8 | 2 | 19 | 10 | 6 | 11 | 18 | 7 |
| Digestive system (42–54) | 5,497 | 171 (3.7) | 48 | 16 | 21 | 14 | 21 | 27 | 14 | 10 |
| Urinary system (55–59) | 2,839 | 105 (5.6) | 5 | 52 | 20 | 3 | 18 | 1 | 3 | 3 |
| Musculoskeletal system (76–84) | 2,051 | 25 (2.7) | 7 | 4 | 3 | 6 | 2 | 1 | 2 | 0 |

[a]Including ventilator-associated pneumonia. *BSI* bloodstream infection. ICD-9-CM (International Classification of Diseases, 9th Revision, Clinical Modification), GI (gastrointestinal infection), LRI (lower respiratory tract infection), NI (nosocomial infection), PD (patient-day), SSI (surgical site infection), URI (upper respiratory tract infection), UTI (urinary tract infection).

## Discussion

In this retrospective cohort of pediatric inpatients undergoing invasive operations, older age, higher WAZ, longer ICU stay, preoperative EN support, antibiotic prophylaxis, and higher hemoglobin level were associated with delayed occurrence of postoperative NIs, while longer preoperative hospitalization, longer operative duration, and higher ASA score might accelerate the occurrence of the postoperative NIs. Risk stratification based on these factors in the same cohort was moderately accurate.

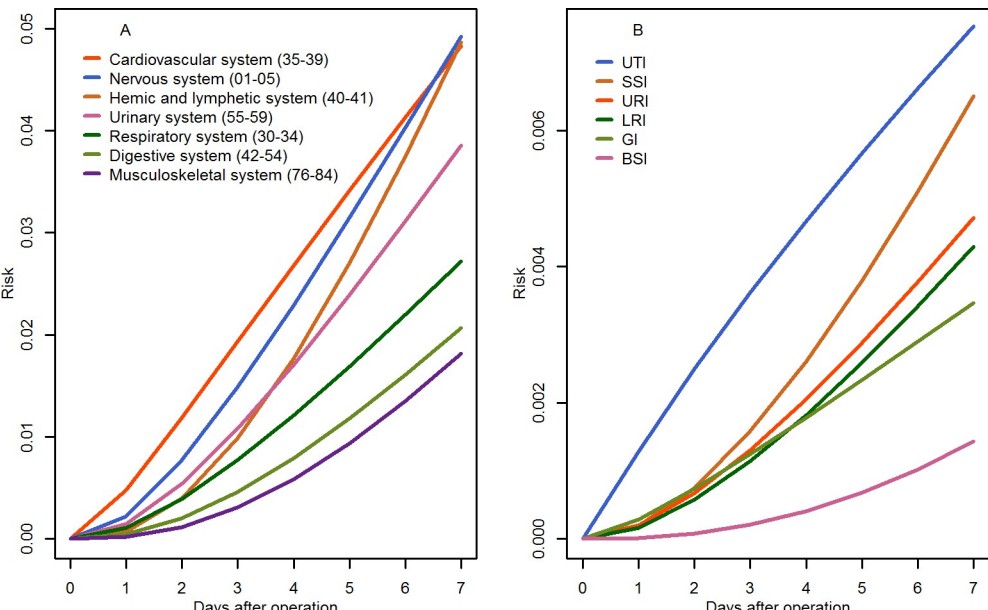

**Fig 1.** Risk of developing NIs in the first postoperative week, stratified by operative site (plot A) and by infection site (plot B). BSI (bloodstream infection), GI (gastrointestinal infection), LRI (lower respiratory tract infection), NI (nosocomial infection), SSI (surgical site infection), URI (upper respiratory tract infection, (UTI) urinary tract infection.

**Table 3. Associations between baseline factors and the development of postoperative NIs in multivariable AFT log-normal models, the Guangzhou Women and Children's Medical Center, 2016–2018.**

| | Full model[a] | | | Reduced model[a] | | |
|---|---|---|---|---|---|---|
| | TR | 95% CI | *P* | TR | 95% CI | *P* |
| Sex, male *vs.* female | 1.07 | 0.94–1.21 | 0.29 | | | |
| Age, per year increase | 1.07 | 1.04–1.10 | < 0.01 | 1.07 | 1.04–1.10 | < 0.01 |
| WAZ | | | | | | |
| Indicator, yes (WAZ = 0) *vs.* NA/missing | 1.31 | 0.91–1.89 | 0.14 | 1.31 | 0.91–1.88 | 0.14 |
| Per unit increase | 1.05 | 1.00–1.09 | 0.05 | 1.05 | 1.00–1.09 | 0.04 |
| Preoperative hospitalization | | | | | | |
| Per day increase | 0.97 | 0.96–0.98 | < 0.01 | 0.97 | 0.96–0.98 | < 0.01 |
| Preoperative ICU stay | | | | | | |
| Indicator, yes (days = 1) *vs.* no | 1.03 | 0.89–1.18 | 0.69 | 1.04 | 0.90–1.20 | 0.59 |
| Per day increase | 1.03 | 1.01–1.04 | < 0.01 | 1.03 | 1.01–1.04 | < 0.01 |
| Preoperative EN, yes *vs.* no | 1.22 | 1.04–1.43 | 0.01 | 1.24 | 1.06–1.45 | 0.01 |
| Preoperative PN, yes *vs.* no | 1.08 | 0.89–1.31 | 0.45 | | | |
| Antibiotic prophylaxis | | | | | | |
| Yes *vs.* no | 1.43 | 1.23–1.66 | < 0.01 | 1.42 | 1.23–1.65 | <0.01 |
| Preoperative blood test | | | | | | |
| Indicator, yes (hemoglobin = 100, WBC = 10) *vs.* no | 0.82 | 0.67–1.00 | 0.04 | 0.82 | 0.68–1.00 | 0.05 |
| Hemoglobin, per 5 g/L increase | 1.03 | 1.02–1.05 | < 0.01 | 1.03 | 1.02–1.05 | < 0.01 |
| WBC, per $5\times10^9$/L increase | 0.99 | 0.97–1.00 | 0.10 | 0.99 | 0.97–1.00 | 0.10 |
| Surgical implantation, yes *vs.* no | 0.89 | 0.74–1.07 | 0.22 | | | |
| Operative duration | | | | | | |
| Indicator, yes (operative duration = 1 hour) *vs.*missing | 2.01 | 1.50–2.69 | < 0.01 | 2.01 | 1.51–2.69 | < 0.01 |
| Per hour increase | 0.91 | 0.87–0.95 | < 0.01 | 0.90 | 0.86–0.94 | < 0.01 |
| ASA score | | | | | | |
| Indicator, yes (ASA score = I) *vs.* missing | 0.67 | 0.47–0.95 | 0.02 | 0.67 | 0.47–0.96 | 0.03 |
| ASA score II *vs.* I | 0.98 | 0.79–1.20 | 0.81 | 0.98 | 0.79–1.20 | 0.82 |
| ASA score ≥III *vs.* I | 0.79 | 0.63–1.00 | 0.06 | 0.79 | 0.62–1.00 | 0.05 |
| SWC | | | | | | |
| Indicator, yes (SWC = clean) *vs.* no | 1.09 | 0.93–1.27 | 0.30 | | | |
| Clean-contaminated *vs.* clean | 0.96 | 0.74–1.24 | 0.75 | | | |
| Contaminated *vs.* clean | 0.87 | 0.59–1.30 | 0.50 | | | |

[a]Both models were adjusted for operative site (ICD-9-CM code). AFT (accelerated failure time model), ASA (American Society of Anesthesiologists), EN (enteral nutrition), ICD-9-CM (International Classification of Diseases, 9[th] Revision, Clinical Modification), ICU (intensive care unit), NA (not applicable), PN (parenteral nutrition), SWC (surgical wound classification), TR (time ratio), WAZ (weight-for-age z-score), WBC (white blood cell).

In this study, NI rate following hemic and lymphatic operations was the highest, probably due to immunodeficiency commonly seen in patients with hematologic diseases. In two studies including one in pediatric surgical patients, the most frequent NI types were SSI [9, 13], -consistent with our results. In the present study, however, the risk of developing UTIs within the first postoperative week was higher than the risks of developing other types of NIs, suggesting that the pathogens causing UTIs might have a relatively short incubation period.

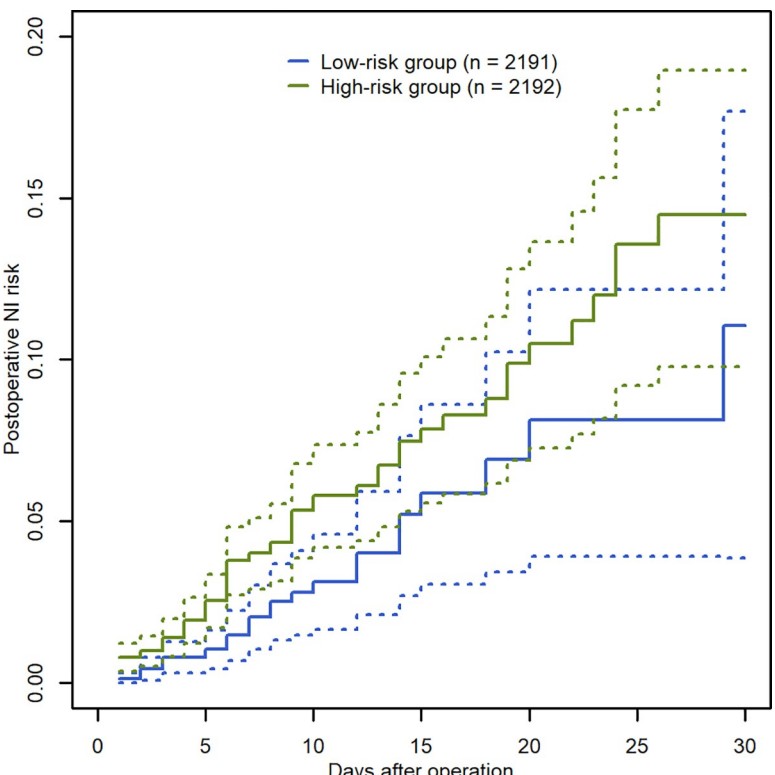

**Fig 2. Postoperative NI risk in the low-risk group (2,192 patients including 21 NIs) and the high-risk group (2,191 patients including 79 NIs), stratified using the median of the risk score derived from the reduced AFT model[a].** [a]Analysis was done in an independent data set of patients undergoing invasive operation between January and May, 2019 (n = 4,383). NI (nosocomial infection).

Malnutrition increases the risk of infection by impairing the immune system [14, 15], accounting for the inverse association between WAZ and postoperative NI risk in this study. Increased NI risk in relation to malnutrition has been reported in prior studies, where malnutrition was defined using different measures such as low birth weight and low body mass index [16, 17]. As a biomarker of malnutrition, low hemoglobin level may compromise non-specific immunity and increase susceptibility to infection [18], which explains its inverse association with postoperative NI risk in our cohort.

For the reference patient we defined, we found that one-day increase in ICU stay before operation was associated with a delay of approximately 2 days in the occurrence of postoperative NIs: this was likely to result from the benefits of intensive care, such as close observation and nutrition support. Previous studies reported a positive association between length of ICU stay and NI risk among ICU patients [19, 20], which however was not surprising given the fact that the risk of developing NIs in ICU always accrues as the ICU stay extends. Furthermore, those studies were flawed by reverse causality (i.e. occurrence of NIs in ICU extends ICU stay) and their problematic use of logistic regression to analyze time-to-event data.

In our cohort, longer preoperative hospitalization was associated with an increased postoperative NI risk: one-week longer preoperative hospital stay would accelerate the occurrence of postoperative NIs by 11 days for the reference patient, in line with a previous study in adult surgical patients [21]. Length of preoperative hospitalization is largely a proxy of the severity and complexity of the underlying disease. Moreover, prolonged preoperative hospitalization

means extended exposure to pathogens and enhanced possibility of developing hospital-acquired malnutrition [22, 23]: both increase patients' susceptibility to infections.

Current evidence regarding the association between EN and NI risk is inconclusive [24, 25]. Thus the inverse association between preoperative EN and postoperative NI risk in our cohort warrants confirmation by future studies. EN may cause NIs via contaminated feedings and/or tracheal colonization of gastric organisms; however, it may also reduce infectious complications by maintaining gut structure [26]. It has been well proved that PN increases the risk of BSI because of CVC involvement [27, 28]. In the present study, only a small fraction of PN (<2%) was administered via CVC, which might partly explain the low incidence of BSIs in the present study as well as the null association between PN and the overall NI risk.

Reduced SSI rates due to antibiotic prophylaxis has been reported [29], supporting our finding that antibiotic prophylaxis might delay the occurrence of postoperative NIs by more than 20 days. However, increasing evidence also suggests that antibiotic prophylaxis will cause unnecessary side effects if it is given without indication [30]. In our cohort, antibiotic prophylaxis was common, but lack of detailed data did not allow us to define antibiotic prophylaxis following the established protocols [31, 32] or to determine its necessity and appropriateness in terms of type, timing, and dosage.

The present study supports the finding of a prior study that higher ASA scores might raise the risks of specific NI types including SSI, UTI, and nosocomial pneumonia [9]. Our results also confirm the existing evidence that prolonged operative duration might increase the risks of postoperative SSIs and other complications [33]. As for SWC, the present study only had a low number of patients with contaminated wounds and had none with dirty wounds. It has also been suggested that surgical wounds might be substantially misclassified in clinical practice [34], making it even more difficult to compare our result with those from previous studies, where a positive association between SWC and the overall postoperative NI risk might exist given the increased SSI risk among patients with contaminated or dirty wounds [35, 36].

Epidemiologic studies on postoperative NI incidence and its risk factors are inadequate, especially for pediatric patients; therefore, our findings would enrich the existing knowledge and would inform healthcare providers of a patient's risk of developing postoperative NIs even before invasive operations. As suggested by our results, a risk score based on basic preoperative and operation-related risk factors might lead to moderately accurate risk stratification. However, using such a risk score would be rewarded with early identification of vulnerable patients, timely prevention, and cautious selection of the postoperative treatment or procedures that might further increase the NI risk.

The strengths of the present study include its relatively large sample size, a variety of possible risk factors to be considered, and proper statistical analysis. From a methodological perspective, despite the largely consistent results, the AFT model was superior to the Cox model in our case, given that the proportional hazards hypothesis could be legitimately violated: for example, the effect of antibiotic prophylaxis will diminish rather than remaining constant over time. Several limitations of the present study should be noted. First, like many other hospital-based studies, this study was subject to selection bias in particular the referral bias, which might limit the generalizability of our results. Second, we were concerned that using ICD-9-CM code at its lowest level of specificity might not be sufficient to control for the invasiveness of the operation and the severity of the underlying disease. Third, several putative risk factors for NIs were either precluded from our analysis or could not be studied thoroughly due to lack of data, such as use of antibiotics and immunosuppressants. Finally, we could not include the NIs that occurred after discharge as no systematic post-hospital follow-up was performed to collect the data.

## Conclusions

Our findings suggest that pediatric patients undergoing invasive operations and at high risk of developing postoperative NIs are likely to be identified using basic preoperative and operation-related risk factors, which together might lead to moderately accurate risk stratification but still provide valuable information to guide early and judicious prevention.

## Supporting information

**S1 File.**
(DOC)

## Author Contributions

**Conceptualization:** Kuanrong Li, Yanqin Cui, Huimin Xia, Xin Sun, Xingrong Song, Huiying Liang.

**Data curation:** Kuanrong Li, Xiaojun Li, Wenyue Si.

**Formal analysis:** Kuanrong Li.

**Methodology:** Kuanrong Li, Wenyue Si, Huiying Liang.

**Project administration:** Wenyue Si, Huiying Liang.

**Resources:** Wenyue Si, Yanqin Cui, Huimin Xia, Xin Sun, Xingrong Song, Huiying Liang.

**Software:** Kuanrong Li, Xiaojun Li, Wenyue Si.

**Supervision:** Yanqin Cui, Huimin Xia, Xin Sun, Huiying Liang.

**Writing – original draft:** Kuanrong Li, Huimin Xia, Xingrong Song, Huiying Liang.

**Writing – review & editing:** Kuanrong Li, Xiaojun Li, Wenyue Si, Yanqin Cui, Huimin Xia, Xin Sun, Xingrong Song, Huiying Liang.

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
