## [Decision Letter · Decision Letter 0]

21 Aug 2019

PONE-D-19-18261

Preoperative and operation-related risk factors for postoperative nosocomial infections in pediatric patients: a retrospective cohort study

PLOS ONE

Dear Dr. Liang,

Thank you for submitting your manuscript to PLOS ONE. After careful consideration, we feel that it has merit but does not fully meet PLOS ONE’s publication criteria as it currently stands. Therefore, we invite you to submit a revised version of the manuscript that addresses the points raised during the review process.

We would appreciate receiving your revised manuscript by Oct 05 2019 11:59PM. To enhance the reproducibility of your results, we recommend that if applicable you deposit your laboratory protocols in protocols.io, where a protocol can be assigned its own identifier (DOI) such that it can be cited independently in the future. For instructions see: http://journals.plos.org/plosone/s/submission-guidelines#loc-laboratory-protocols

We look forward to receiving your revised manuscript.

Kind regards,

Agnieszka Rynda-Apple, Ph.D.

Academic Editor

PLOS ONE

Journal Requirements:

2. Please note that all PLOS journals ask authors to adhere to our policies for sharing of data and materials: https://journals.plos.org/plosone/s/data-availability. According to PLOS ONE’s Data Availability policy, we require that the minimal dataset underlying results reported in the submission must be made immediately and freely available at the time of publication. As such, please remove any instances of 'unpublished data' or 'data not shown' in your manuscript and replace these with either the relevant data (in the form of additional figures, tables or descriptive text, as appropriate), a citation to where the data can be found, or remove altogether any statements supported by data not presented in the manuscript.

Additional Editor Comments:

Dear Dr. Liang,

We have completed our review of your manuscript PONE-D-19-18261 entitled "Preoperative and operation-related risk factors for postoperative nosocomial infections in pediatric patients: a retrospective cohort study". While our review indicated that the manuscript is not suitable for publication in PlosONE in its current state, the work itself seems to be sound. With the necessary revisions, this manuscript could be reconsidered. The comments of the reviewers are included at the bottom of this letter. While responding to the comments and criticisms voiced by the reviewers please provide more detailed description of statistical models and analysis used.

Per PlosONE policy the data underlying the findings of the manuscript needs to be fully available; either included as supplementary information or deposited to the public repository. If there are any restrictions on publicly sharing data, those need to be specified.

Yours sincerely,

Agnieszka Rynda-Apple

Academic Editor

PLOS ONE

Reviewers' comments:

Reviewer's Responses to Questions

**Comments to the Author**

1. Is the manuscript technically sound, and do the data support the conclusions?

Reviewer #1: Yes

Reviewer #2: Partly

Reviewer #3: Partly

2. Has the statistical analysis been performed appropriately and rigorously? 

Reviewer #1: Yes

Reviewer #2: Yes

Reviewer #3: I Don't Know

3. Have the authors made all data underlying the findings in their manuscript fully available?

Reviewer #1: No

Reviewer #2: Yes

Reviewer #3: No

4. Is the manuscript presented in an intelligible fashion and written in standard English?

Reviewer #1: Yes

Reviewer #2: Yes

Reviewer #3: Yes

5. Review Comments to the Author

Reviewer #1: The paper is well written, methods appear appropriate, and the results provide useful information on risk factors for post-operative infection following paediatric surgery. The reference list needs tidying for consistent formatting.

Reviewer #2: This manuscript provides a retrospective analysis of nosocomial infections in pediatric patients undergoing invasive operations.

Major Comments:

- The statistical inference lacks scientific context beyond stating statistical significance. The American Statistical Association's statement on p-values highlights this point: \\emph{A p-value, or statistical significance, does not measure the size of an effect or the importance of a result.} For instance, line 269 states ``We found that longer preoperative ICU stay was associated with delayed occurrence of postoperative NIs:...`` How influential was this in terms of the occurrence probability? The entire discussion lacks this effect size discussion.

- L 109 - 111. What are the implications of only the first invasive operation / postoperative NI being analyzed? Does this result in any bias? Is it expected that follow up surgeries would have different rates of NI occurrence? My understanding is that if the first invasive operation did not result in a NI, then this data point would be included and all later invasive operations would be excluded. Is this correct?

- L 139 - 140. Be specific about how the variables were treated as `missing'. Missing data is a loaded word with a whole subfield of statistics devoted to it. How does the missing data and associated binary indicator impact the statistical models?

- L 148 - 149. How do the standard discharge rates (days after surgery) differ between the surgery types? Figure 1 shows the risk up to 7 days post operation. How large would the sample sizes be for 7 - days post operation for all surgery types? I'm curious if the analysis with incidence rate per patient days is a good comparison if many (most!) patients of a certain surgery type are released from the hospital shortly after surgery. How is this treated in the statistical models? Is this type of analysis standard, if so include supporting evidence/citations, otherwise, would an analysis of likelihood of experiencing an NI for a given surgery be appropriate?

- How are the ICD-9-CM codes used in the statistical models? Table 2 only states both models were adjusted for operative site - how so? Are these intercept values or would a more general model (such as a hierarchical model) that allows different responses to explanatory variables across the groups be more appropriate?

- Line 234. How are the high-risk and low-risk groups defined? I don't see this in the text, but the caption for Figure 2 states, "median of the risk score derived from the reduced AFT model". Why is this grouping useful or interesting? It is not surprising that the high-risk group would have higher NI incidence rates, but this does suggest the overall model is doing something useful.

Minor Comments:

- Use commas to separate five or more numbers, such as 18314 $\\rightarrow$ and 18,314.

- L 80, From prevention perspective $\\rightarrow$ from a prevention perspective.

- L 155 - 159. It is not clear how the block structures works in practice. Do the p-values for all variables in a block need to be under the specified threshold?

- P 9. ``averagely'' could be reworded/replaced.

- Table 1 contains a lot of information. One thing that might help would be lines or something to signify sub-categories (within SWC and ASA score).

- Table 2, typo ``paediatric''

- Table 3. Is the full model necessary?

Reviewer #3: This review focuses on the *statistical aspects* of the submitted manuscript, exclusively.

In essence, the manuscript describes the statistical analysis of a hospital data-set containing pediatric peri-operative data and nosocomial infection status; this reviewer understands the goal of the study as identification of pre-operative and operation-related risk factors for NI.

The authors chose "time to event" as primary variable of interest, where event specifies either NI or discharge. (see major comment, below). The main model is "Accelerated failure time" (AFT). A (more standard, but measuring something slightly different) Cox model was used for comparison (but data only reported in appendix 2). The authors state that non-linear associations of continuous factors were considered, but none retained (if this reviewer understands correctly).

A reduced model was obtained by backward elimination. Identified time-ratios of factors were used to classify patients into low- and high-risk, respectively, with an enrichment of 70% of NI in the high-risk group.

The manuscript is well written, concise, and well-structured. There are a few items where this reviewer feels that the description should be more complete (verbose, but potentially also include a few math-stats expression for precision). Overall, this reviewer believes that with appropriate revisions the manuscript should be published (unless revisions bring major statistical flaws to light).

Comments that the authors should to address in revision:

1) major one: The authors are dealing with a highly censored data set. A majority of patients does not exhibit NI (or not on record, anyways). It is absolutely not clear to this reviewer, how the authors are handling censored events (i.e. distinguish time to event between NI and discharge). In particular since NI patients are in the minority, the main event is discharge---is the AFT model therefore primarily fitting time to discharge? It is this reviewer's strong belief that this "detail" should be described and discussed in much more detail, here. What are the exact assumptions? What does the AFT model look like (a mathematical expression or two would be appreciated)? How "log-normal" are the data (show it?)? How do NI events differ from discharge events? Is the AFT model fit to patients with NI, only, or the entire collection? etc.... Maybe the authors are doing everything correct, already, but from the description it is near impossible to know (or possible reproduce). This is the central element of the study and should be presented with sufficient detail (despite space limitations...).

2) The primary outcome of the AFT model is that certain factors accelerate or decelerate the progression of a patient through progress from operation to event (NI/discharge). How does this relate to NI *risk* (=what the cox model captures)? A description of the exact relationship between these "measures" should be provided.

3) The non-linear fractional polynomial part is dealt with in twice half a sentence --- either this part is important enough to deserve some more details (then some more details on this should be provided) --- or it was just a "stump track" and should parenthetically be noted as such.

4) it is this reviewer's understanding that after model fitting, significant factors were aggregated through non-linear averaging (median) to produce a "predictive score" indicating low or high NI risk. This reviewer would appreciate some more details in the description, as this is a key contribution.

5) One of the presented results is that after risk classification, the high-risk patients exhibit twice as big empirical risk for NI compared to low-risk group. Immediately resulting thereof, the high-risk group accounts for a bit over 2/3s of the NI occurrences. Can the authors compare this figure against other models that are already "on the market"? (a random grouping into equally large groups would result in a roughly 50% split; how much "better than random" is the observed 70%?)

6) Is there a compact way to graphically visualize the content of table 3 (and possibly appendix 2)? TR and HR CI, point estimates and p-values could be shown as "boxplot"-like bars relative to the neutral 1-axis. In this way, significant and large impacts will be more intuitively identifiable from the large crowd.

7) Discussion: the results are presented as "delayed occurrence" versus "enhanced postoperative risk". Again, in the light of point 1 and 2, it is not entirely clear to this reviewer what is talked about: delayed occurrence of NI (or discharge?!)? How is post-operative risk quantified, here? (= accelerated occurrence of NI/discharge?) A priori delayed occurrence and risk are two different things that can not immediately be compared. The description should be a bit more verbose and complete, here.

8) line 256: consisting => consistent

9) This reviewer is afraid that the list of possible (likely and relevant) confounders should be longer.

10) Cross-validation: since the data are used twice---first to learn the AFT model parameters, second to stratify into risk-groups---it would be indicated to use separate parts of the data set for these tasks, to control "overfitting". How good is the risk-stratification outcome when applied on 50% of the data that was NOT used to fit the AFT model used to identify risk factors? The differences may be marginal, but it would be prudent to check.

11) one in ten rule (and variants thereof): given 847 un-censored events how many predictor parameters can reasonably be expected to be fit? The authors might want to include a note on this.

6. PLOS authors have the option to publish the peer review history of their article (what does this mean?). If published, this will include your full peer review and any attached files.

Reviewer #1: No

Reviewer #2: No

Reviewer #3: No

---

## [Author Response · Author response to Decision Letter 0]

10 Oct 2019

Please refer to attached the rebuttal letter

---

## [Decision Letter · Decision Letter 1]

8 Nov 2019

Preoperative and operation-related risk factors for postoperative nosocomial infections in pediatric patients: a retrospective cohort study

PONE-D-19-18261R1

Dear Dr. Liang,

We are pleased to inform you that your manuscript has been judged scientifically suitable for publication and will be formally accepted for publication once it complies with all outstanding technical requirements.

With kind regards,

Agnieszka Rynda-Apple, Ph.D.

Academic Editor

PLOS ONE

Reviewers' comments:

Reviewer's Responses to Questions

**Comments to the Author**

1. If the authors have adequately addressed your comments raised in a previous round of review and you feel that this manuscript is now acceptable for publication, you may indicate that here to bypass the “Comments to the Author” section, enter your conflict of interest statement in the “Confidential to Editor” section, and submit your "Accept" recommendation.

Reviewer #2: All comments have been addressed

2. Is the manuscript technically sound, and do the data support the conclusions?

Reviewer #2: Yes

3. Has the statistical analysis been performed appropriately and rigorously? 

Reviewer #2: Yes

4. Have the authors made all data underlying the findings in their manuscript fully available?

Reviewer #2: Yes

5. Is the manuscript presented in an intelligible fashion and written in standard English?

Reviewer #2: Yes

6. Review Comments to the Author

Reviewer #2: (No Response)

7. PLOS authors have the option to publish the peer review history of their article (what does this mean?). If published, this will include your full peer review and any attached files.

Reviewer #2: No

---

## [Editor Report · Acceptance letter]

13 Dec 2019

PONE-D-19-18261R1 

Preoperative and operation-related risk factors for postoperative nosocomial infections in pediatric patients: a retrospective cohort study 

Dear Dr. Liang:

I am pleased to inform you that your manuscript has been deemed suitable for publication in PLOS ONE. Congratulations! Your manuscript is now with our production department. 

With kind regards,

on behalf of

Dr. Agnieszka Rynda-Apple 

Academic Editor

PLOS ONE